# The Epidermal Transcriptome Analysis of a Novel c.639_642dup *LORICRIN* Variant-Delineation of the Loricrin Keratoderma Pathology

**DOI:** 10.3390/ijms24119459

**Published:** 2023-05-29

**Authors:** Katarzyna Wertheim-Tysarowska, Katarzyna Osipowicz, Bartłomiej Gielniewski, Bartosz Wojtaś, Alicja Szabelska-Beręsewicz, Joanna Zyprych-Walczak, Adriana Mika, Andrzej Tysarowski, Katarzyna Duk, Agnieszka Magdalena Rygiel, Katarzyna Niepokój, Katarzyna Woźniak, Cezary Kowalewski, Jolanta Wierzba, Aleksandra Jezela-Stanek

**Affiliations:** 1Department of Medical Genetics, Institute of Mother and Child, 01-211 Warsaw, Poland; 2Department of Dermatology, Immunodermatology and Venereology, Medical University of Warsaw, 02-008 Warsaw, Poland; 3Laboratory of Molecular Neurobiology, Nencki Institute of Experimental Biology, 02-093 Warsaw, Poland; 4Department of Mathematical and Statistical Methods, Poznań University of Life Sciences, 60-637 Poznań, Poland; 5Department of Pharmaceutical Biochemistry, Medical University of Gdansk, 80-211 Gdansk, Poland; 6Molecular and Translational Oncology Department and Cancer Molecular and Genetic Diagnostics Department, Maria Sklodowska-Curie National Research Institute of Oncology, 02-781 Warsaw, Poland; 7Department of Paediatrics, Haematology and Oncology, Department of General Nursery, Medical University of Gdansk, 80-211 Gdansk, Poland; 8Department of Genetics and Clinical Immunology, National Institute of Tuberculosis and Lung Diseases, 01-138 Warsaw, Poland

**Keywords:** loricrin, loricrin keratoderma, transcriptome, mutation

## Abstract

Loricrin keratoderma (LK) is a rare autosomal dominant genodermatosis caused by *LORICRIN* gene mutations. The pathogenesis of the disease is not yet fully understood. So far, only 10 pathogenic variants in *LORICRIN* have been described, with all of them but one being deletions or insertions. The significance of rare nonsense variants remains unclear. Furthermore, no data regarding the RNA expression in affected patients are available. The aim of this study is to describe the two variants in the *LORICRIN* gene found in two distinct families: the novel pathogenic variant c.639_642dup and a rare c.10C > T (p.Gln4Ter) of unknown significance. We also present the results of the transcriptome analysis of the lesional loricrin keratoderma epidermis of a patient with c.639_642dup. We show that in the LK lesion, the genes associated with epidermis development and keratocyte differentiation are upregulated, while genes engaged in cell adhesion, differentiation developmental processes, ion homeostasis and transport, signaling and cell communication are downregulated. In the context of the p.Gln4Ter clinical significance evaluation, we provide data indicating that *LORICRIN* haploinsufficiency has no skin consequences. Our results give further insight into the pathogenesis of LK, which may have therapeutic implications in the future and important significance in the context of genetic counseling.

## 1. Background

Loricrin keratoderma (LK, MIM 604117, Vohwinkel syndrome with ichthyosis, VS) is a rare autosomal dominant genodermatosis caused by pathogenic variants in the *LORICRIN* gene. *LORICRIN* encodes one of the key proteins conferring the insolubility, mechanical resistance and impermeability of the epidermal barrier [1] 

Hydrophobic and insoluble loricrin is expressed in the orthokeratinizing epithelia, except for internal ones. In the skin, its synthesis occurs in the upper layer of epidermis-*stratum granulosum* (SG). Loricrin is involved in cytoskeleton stabilization forming crosslinks within and between the proteins, and in the formation of the cornified cell envelope (CE), being the most abundant (commonly > 70%) protein there [2,3]. 

Consequently, the clinical symptoms of LK patients are related to the skin surface and comprise the following: ichthyosis; palmoplantar keratoderma (PPK), often with honeycomb pattern, pseudoainhum and/or amputation; knuckle pads; and collodion membrane at birth [4]. Of note, the clinical symptoms are heterogenous and may differ even among relatives [5]. 

The data concerning LK are limited. Only 10 pathogenic variants in 21 affected families (overall 106 patients) were described in the literature so far [4]. Moreover, all of them, apart from one substitution, are deletions/insertions. The only pathogenic missense variant known so far was identified as causative in late-onset loricrin keratoderma [6]. The clinical significance of the other variant types remains questionable.

The aim of the study is to describe two variants in the *LORICRIN* gene that were found during diagnostic procedures of cornification disorders. The first one, c.639_642dup (p.Thr215GlyfsTer122), is a novel pathogenic variant detected in a family with autosomal dominant hyperkeratosis, for which we also present the results of a transcriptomic analysis. This is the first transcriptomic analysis of a loricrin keratoderma lesion. Another variant, the rare c.10C > T (p.Gln4Ter), was detected in the other family as a secondary finding of unknown significance. Considering the highly limited data on the clinical significance of *LORICRIN* premature stop codon (PTC) variants, we provide data showing that p.Gln4Ter leading to a premature stop codon has no skin consequences. This has important significance in the context of genetic counseling. 

## 2. Experimental Design

All patients gave informed consent to participate in the study. 

### Patients

Family 1: The family (two daughters and their father, Figure 1A) was referred to genetic counselling because of hyperkeratosis of the palms and soles and a clinical diagnosis of ichthyosis. The clinical symptoms were manifested by ichthyosiform dermatosis, diffuse generalized. In the girls, the symptoms were noted at birth. Then, palmoplantar keratoderma occurred at the age of 2–3 months. The course of the disease varied, with occasional exacerbations. The improvement was noted after the use of emollients. Occasionally, the transgradient extension of hyperkeratosis onto the wrists and on the bends of the elbows and knees was present, pseudoainhum was not observed and, according to the patient, was also absent in the other affected family members. The honeycomb pattern of PPK was negative during clinical evaluation and, according to the mother, had not been observed before. The keratoderma was neither painful nor inflammatory.

Family 2: The proband was a girl born from an uneventful pregnancy at 38 weeks of gestation (birth weight 2880 g, Apgar score: 7). She had clinical recognition of autosomal recessive congenital ichthyosis (ARCI), due to a homozygous pathogenic variant c.1562A>G (p.Tyr521Cys) in *ALOX12B*. The symptoms of ARCI were typical (collodion baby; later-in-life dryness of the face skin and, less intensive, of the whole body; fingers and toes contractures; erythema; stiff and cracking skin of the hands and feet; slight psoriasis lesions on the knees and elbows), no nail and hair disturbances were present and the teeth appeared normally, though a slight yellow discoloration of permanent teeth was observed (Figure 2). 

The *LORICRIN* variant p.Gln4Ter was detected as a secondary finding during a molecular test. A segregation analysis has shown that the variant was inherited from the patient’s father. A dermatological evaluation of the father did not reveal any skin symptoms at the age of 41; only dystrophic nails were present and massive caries (currently with upper teeth dentures) from the age of 20. 

## 3. Results

### 3.1. Genetic Analysis

We identified a novel variant in the *LORICRIN* gene: c.639_642dup (p.Thr215GlyfsTer122) in family 1 (Figure 1A). The pathogenicity status was scored as likely pathogenic (LP) according to the American College of Medical Genetics (ACMG) classification. Importantly, similarly to other *LORICRIN* pathogenic variants reported so far, the c.639_642dup caused delayed translation termination and introduced an arginine and leucine reach sequence. Thus, the diagnosis of loricrin keratoderma was established. 

### 3.2. Transcriptome Analysis of the Probant vs. Control

The 15,210 genes with distinct ensemble identification (ID) and more than five counts in each sample were detected. Considering the fact that the data analysis was largely limited and included a single patient vs. single control analysis, we highly strengthened the differentially expressed genes (DEG) parameters to the absolute value of logarithm fold change (|logFC|) > 3 and logarithm of counts per million reads (logCPM) > 1, resulting in 1722 genes. Among them, 276 genes were upregulated (logFC between 3.05 and 12.7) and 1445 downregulated (logFC between −3.0 and −14.45). However, only in 10 and 53 genes, respectively, was the statistical difference significant (*p*-value < 0.005) (Table 1). With respect to ontology, genes-encoding proteins involved in epidermis development and keratinocyte differentiation were mainly upregulated. In turn, those engaged in, i.e., cell adhesion, developmental processes and anatomical structure morphogenesis, cellular ion homeostasis and transport, cell differentiation, regulation of signaling and cell communication were downregulated (Table 2). 

## 4. Discussion

In this study, we described the novel *LORICRIN* gene pathogenic variant: c.639_642dup with the first transcriptome analysis of lesional loricrin keratoderma epidermis, and a rare p.Gln4Ter variant in the same gene, as evidence that the haploinsufficiency of loricrin does not cause skin symptoms of LK. 

The in silico prediction showed that the consequence of c.639_642dup on the protein level is a generation of a sequence rich in basic amino acids, mainly arginine. It has already been proven that all the other known frameshifts in the C-part of the loricrin also lead to the formation of arginine-rich regions generating nuclear localization signals (NLS) [7]. Indeed, such loricrin derivative mutated proteins were found to be deposited in the nucleus and distort epidermal differentiation [5,8]. This was also observed in a mouse model of LK, where mutated loricrin was almost exclusively present in the nucleus. This, in fact, was further proven to be an LK-causative factor. It was also shown that the LK phenotype of transgenic mice was more severe in the absence of wild type loricrin [8]. 

Next-generation sequencing technologies (NGS) enabled robust progress in the genetics of the disorders of cornification. While DNA sequencing has already revealed a plethora of disease-causing variants, showing great heterogeneity in the molecular basis of these diseases, RNA sequencing data from these patients are rather limited. Nevertheless, a few studies have already shown that transcriptome analyses may be crucial for obtaining deeper insight into the pathophysiology of the cornification disorders. However, according to our knowledge, no data on the gene expression in loricrin keratoderma patients are available, probably due to the rarity of this disorder. 

Herein, we showed the results of the transcriptome analysis performed using mRNA isolated from the lesion epidermis of the patient with heterozygous novel variant c.639_642dup. In total, 1722 genes were differentially expressed, of which 276 genes were upregulated and 1445 downregulated. However, only 10 and 53 genes reached statistical significance, respectively. 

The *HRNR*-encoding hornein was the most upregulated gene. This gene is located on chromosome 1q21 within the human epidermal differentiation complex (EDC). Hornein belongs to S100 fused-type proteins (SFTPs) and is involved in the cornified epithelium formation [9]. Furthermore, hornein has an antimicrobial activity as the source of cationic intrinsically disordered antimicrobial peptides (CIDAMPs) [10,11]. It has previously been shown that *HRNR* mRNA expression increased transiently in cultured human epidermal keratinocytes during Ca^2+^-dependent differentiation [12]. Of note, Rice et al. and Kim et al. have shown that in healthy people, the *HRNR* is preferably expressed in palmoplantar skin compared to other regions [13,14].

So far, the *HRNR* gene was mainly analyzed in the context of the other skin diseases: psoriasis and atopic dermatitis (AD), where barrier defects occur as well, but due to distinct immunogenetic factors. The *HRNR* transcripts were detected in regenerating human skin after wounding in the periphery regions of psoriatic lesions [15]. Moreover, the hornein immunoreactivity in the lesions, but not in the healthy skin, of psoriasis and atopic dermatitis patients was also diminished in another study [12]. Furthermore, Henry et al. showed that the expression was lower also in the healthy skin of AD patients. The authors demonstrated that hornein is a component in a cornified envelope (CE) and suggested that it plays a role in the alterations in the CE and in the abnormality of the AD epidermal barrier [16]. 

Just recently, Makino et al. checked the HRNR expression by immunostaining in skin lesions from patients affected by hyperkeratosis-associated diseases (ichthyosis vulgaris, epidermolytic ichthyosis (EI), Darier’s disease, lichen planus, pustulosis et plantaris, actinic keratosis and seborrheic keratosis). The increased expression was detected in lichen planus and pustulosis et plantaris, followed by an irregular signal pattern in epidermolytic ichthyosis and actinic keratosis. In the remaining diseases (ichthyosis vulgaris, Darier’s disease and seborrheic keratosis), the expression was decreased. Thus, in light of our results and those mentioned above, further studies are needed to evaluate the hornein involvement in epidermal barrier restoration [17].

Among the other top 10 upregulated genes, we detected a few more associated with barrier formation: *LCE3D* (late cornified envelope protein 3d), *KRT9* (keratin 9) and *CDSN* (corneodesmosin). Those genes were also found to be upregulated in the other types of ichthyoses [18]. Specifically, *LCE3D* was also found to be upregulated in the other diseases with keratoderma: Pachyonychia Congenita and Curth-Macklin ichthyosis [19,20].

Due to the fact that our analysis consisted of only one patient and one control, the statistical analysis was very limited. Therefore, we also focused on the genes that had logFC over 3.0 or below −3.0 and logCPM > 1, irrespective of the p-value. In this group, several others had induced expression as well, including the IL-17/TNF-α-associated molecules *IL36G* and *S100A9*. Of note, previous studies also showed that the Th17 pathway is induced in various forms of ichthyosis, which proves that in terms of immune response, ichthyoses resemble psoriasis. Hence, novel therapies using IL-17 may be deliberated in the future [18,21]. Overall, among the 276 upregulated genes, those associated with epithelium development, keratocyte differentiation and keratinization were the most represented.

It has been shown that apart from some commonly expressed genes, different ichthyoses vary in terms of gene expression. This is reflected even by the numbers of DEG (patient’s lesions vs. control) in different disorders. Malik et al. showed that in the Netherton syndrome patient, the number was relatively low: 63 upregulated and 33 downregulated DEGs comparing to epidermolytic ichthyosis (EI), where the number of DEGs was 223 and 150, respectively. Furthermore, Kim et al. identified lipid metabolism and barrier junction genes to be downregulated in four common ichthyosis types, which were less pronounced in EI [21]. Furthermore, Malik et al. proved that the expression of lipid metabolism genes was diminished in lamellar ichthyosis (LI) patients, but not, or to a lesser extent, in EI [18]. This phenomenon may result from the distinct molecular basis of those disorders: LI is mainly caused by mutations in genes involved in lipid metabolism, while in EI, pathogenic variants in structural keratins 1 and 10 are causative. Finally, when we compared the genes downregulated in our patient with those published by Ortega-Recalde et al. in Curth-Macklin ichthyosis, two were concordantly downregulated (*PHYHIP*, *PAMR1*), while *DCD*, *FABP4*, *PLIN1*, *SCGB1D2*, *SCGB2A2*, *ADIPOQ*, *G0S2*, *KRT19* and *MUCL1*, downregulated in our case, were upregulated in Curth-Macklin ichthyosis. 

Among the mostly downregulated 53 genes, we found a few involved in lipid metabolism, e.g., *PLAAT3*, *FABP4*, *PLIN1/4* and *PLNPLA6*. However, a gene ontology analysis performed in the wider context showed that the majority of 1445 genes detected by us are involved in cell adhesion developmental processes and anatomical structure morphogenesis, cellular ion homeostasis and transport, cell differentiation, regulation of signaling and cell communication. This finding is in line with the molecular pathomechanism of loricrin keratoderma, which, as already mentioned before, comprises the nuclear deposition of mutated loricrin and the dysregulation of keratinocyte differentiation. Of note, once we compared the 53 mostly downregulated genes of our patient with the DEG profile of atopic dermatitis and psoriasis presented by Malik et al., 14 also had a diminished expression in AD and 15 in psoriasis, whereas only a few (1–5) overlapped with other ichthyoses [18]. 

Since the results, to our knowledge, are the first transcriptome analysis of LK lesion and were performed on the one patient only, replicative studies are needed. Nevertheless, our results provide novel insight into the pathogenesis of the disease and may have therapeutic implications in the future. 

Another issue raised by us concerns the clinical significance of nonsense variants in *LORICRIN.* It has been shown that transgenic mice with one copy of the *loricrin* gene are phenotypically normal [22]. However, as far as we can tell, there are no phenotypic descriptions of humans with one functional copy of the *LORICRIN* gene available thus far. Hence, the genetic counseling in such cases may be ambiguous.

In the ClinVar database, one premature stop codon (PTC) variant [NM_000427.3:c.624C > G (p.Tyr208Ter), ID: 1324671] is recorded and is assigned as likely pathogenic. On the contrary, in the SNP database (SNPdb), 13 nonsense variants are recorded, with the frequency ranging from 0 to 0.00004, according to the GnomAD or Kaviar databases. None of the variants were detected in homozygosity and each was classified as VUS according to the ACMG [23] classification. The variant c.10C > T (p.Gln4Ter) detected in family 2 is also recorded in SNPdb and was found in 2 out of 231 412 GnomAD alleles of European, non-Finnish ancestry. 

The proband of family 2 was diagnosed as autosomal recessive congenital ichthyosis ARCI with *ALOX12B* biallelic mutations; therefore, it was impossible to initially correlate the clinical symptoms with the *LORICRIN* genotype. Since we have shown that the c.10C > T (p.Gln4Ter) variant was of paternal origin, the father was clinically evaluated. There was no history of skin involvement, but dystrophic nails and massive carries from the age of 20 were reported. Nail involvement in LK is uncommon and also was not described in knock-out mice models [1,22,24], although loricrin is expressed in the nail proximal fold [25]. Nevertheless, considering the fact that the father of family 2’s history of dystrophic nails was negative, as well as the fact that there were no nail symptoms in the ARCI-affected proband, the nail dystrophy of the father seemed to occur independently. There were also no cases of massive caries among the father’s relatives. Interestingly, previous studies have shown that in murine and human aggressive periodontitis, *LORICRIN* mRNA expression was diminished [26,27]. Therefore, though no skin changes were noted, an open question remains as to whether the presence of the heterozygous PTC variant confers susceptibility to caries. 

In conclusion, our results broaden the knowledge about *LORICRIN* gene variants and their phenotypic significance and give insight into the molecular pathology of loricrin keratoderma lesions. 

## 5. Methods

### 5.1. Molecular Analysis

Genotyping was performed on DNA isolated from blood leukocytes using a Genomic Maxi AX kit (A&A Biotechnology, Gdańsk, Poland). A customized gene panel (NimbleDesign, Roche, Basel, Switzerland) for Mendelian disorders of cornification was performed. (The panel contains coding exons of *AAGAB*, *ABCA12*, *ABHD5*, *ADAM10*, *ALDH3A2*, *ALOX12B*, *ALOXE3*, *AP1S1*, *AQP5*, *CDSN*, *CLDN1*, *CSTA*, *CTSC*, *CYP4F22*, *DSG1*, *DSP*, *EBP*, *ENPP1*, *ERCC2*, *ERCC3*, *FERMT1*, *FLG* [fragment covering amino acids 1-2200], *GJA1*, *GJB2*, *GJB3*, *GJB4*, *GTF2H5*, *HOXC13*, *JUP*, *KANK2*, *KRT1*, *KRT10*, *KRT16*, *KRT17*, *KRT2*, *KRT9*, *LIPN*, *LOR*, *MBTPS2*, *MPLKIP*, *NIPAL4*, *NSDHL*, *PEX7*, *PHYH*, *PKP1*, *PNPLA1*, *POFUT1*, *POGLUT1*, *POMP*, *SERPINB7*, *SLC27A4*, *SLURP1*, *SNAP29*, *SPINK5*, *ST14*, *STS*, *SUMF1*, *TGM1*, *TRPV3*, *VPS33B*.) The libraries were prepared using the KAPA Library Preparation Kit (Roche, Basel, Switzerland) and sequenced on the MiSeq (Illumina, San Diego, CA, USA). The reads were aligned against GRCh38 human genome assembly. The variants were annotated using the following: the SNPdb (NCBI, Bethesda, MD, USA), ExAC, Ensembl, OMIM, GnomAD, ClinVar and HGMD Professional. Varsome was used to evaluate the ACMG [23] score, and IGV to see the bam files. Selected variants were confirmed using Sanger sequencing (primers and PCR settings available upon request). The familial analysis was limited to the Sanger analysis only. 

### 5.2. Skin Biopsy

The transcriptome analysis was performed using RNA isolated from lesional epidermis. The 3 mm skin biopsy from the lesion located on the upper tibia was taken from the LK patient and from the same location of a healthy age-matched male. The biopsies were immediately frozen and kept at −80 °C. The epidermis was mechanically detached from the underlying skin layers in a cryotome prior to RNA isolation.

### 5.3. RNA Sequencing

The samples were mechanically homogenized, and RNA was isolated using an RNeasy Micro Kit (Qiagen, Hilden, Germany). The quality and integrity of total RNA were assessed with an Agilent 2100 Bioanalyzer using an RNA 6000 Pico Kit (Agilent Technologies, Ltd. Santa Clara, CA, USA) In total, polyA enriched RNA libraries were prepared using the QuantSeq 3′ mRNA-Seq Library Prep Kit according to the manufacturer’s protocol (Lexogen GmbH, Vienna, Austria). Briefly, libraries were prepared from 5 ng of total RNA. The first step in the procedure was a first-strand cDNA synthesis using reverse transcription with oligodT primers. Then, all remaining RNA was removed to what was essential for an efficient second-strand synthesis. The second-strand synthesis was performed to generate double-stranded cDNA (dsDNA). It was initiated by a random primer containing an Illumina-compatible linker sequence. The obtained cDNA was purified using magnetic beads to remove all reaction components. cDNA libraries were amplified by PCR using starters provided by a producer. The library evaluation was completed with an Agilent 2100 Bioanalyzer using the Agilent DNA High Sensitivity chip (Agilent Technologies, Ltd., Santa Clara, CA, USA). The mean library size was 220 bp. Libraries were quantified using a Quantus fluorometer and QuantiFluor double-stranded DNA System (Promega, Madison, WI, USA). Libraries were run in the rapid run flow cell and were single-end sequenced (75 bp) on HiSeq 1500 (Illumina, San Diego, CA, USA). 

### 5.4. Statistical Analysis

The quality of sequencing data was firstly checked with the FastQC program [28]. Then, data were mapped to the reference human genome GRCh38 with a star aligner [29]. The calculation of read counts was performed with the HT seq [30]. All genes with very low expression (below 5 counts) across the examined samples were discarded. Due to no replicates for the differential expression analysis, the edgeR method [31], recommended for such an experimental design, was used. We used the value 0.75 as an approximation of the dispersion parameter based on our previous experience with similar data. The gene ontology was performed using the system PipeR package [32]. As important genes, those with an absolute value of log fold change higher than 3 and an abundance of read measured by log counts per million higher than one were chosen. All statistical analyses were carried out using R software v. 4.2.3 [33].

## Figures and Tables

**Figure 1 ijms-24-09459-f001:**
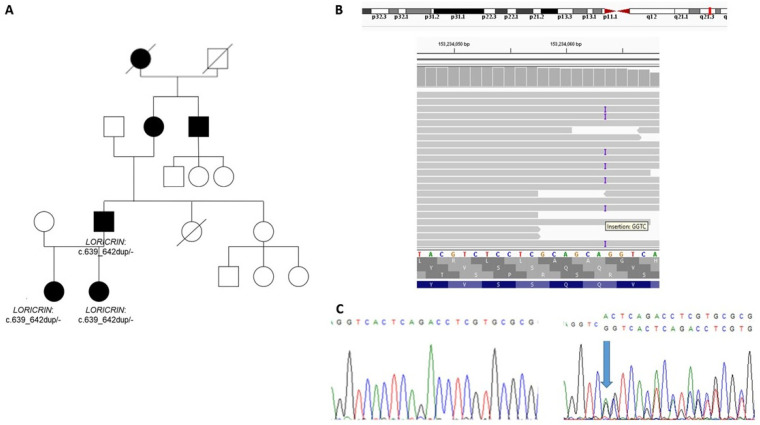
FAMILY 1: (**A**) pedigree; (**B**) results of NGS analysis showing c.639_642dup site in the *LORICRIN* gene, the IGV view; (**C**) confirmatory Sanger sequencing showing fragment of the wild type *LORICRIN* sequence (left one) and the sequence with heterozygous c.639_642dup (right one, marked by an arrow).

**Figure 2 ijms-24-09459-f002:**
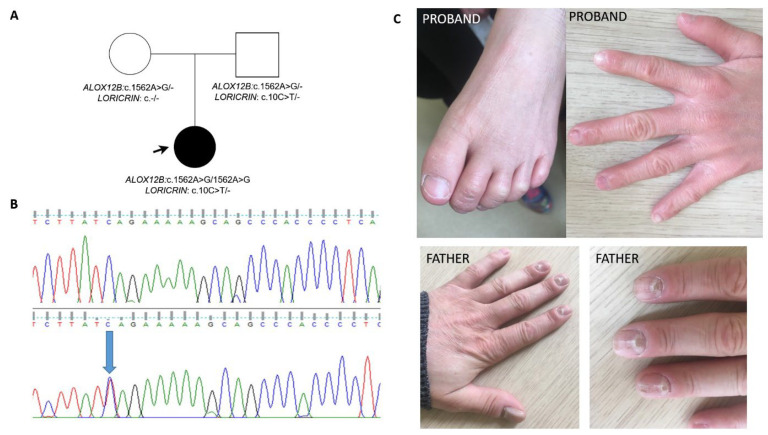
FAMILY 2: (**A**) pedigree; (**B**) confirmatory Sanger sequencing showing fragment of the wild type *LORICRIN* sequence (upper one) and the sequence with heterozygous c.10C > T (p.Gln4Ter, lower one, marked by an arrow); (**C**) the nails and hand of the family 2′s proband (daughter), and of her father.

**Table 1 ijms-24-09459-t001:** The list of the mostly upregulated and downregulated genes expressed in loricrin keratoderma patients.

Symbols	Name	Ensembl ID	logFC	logCPM
upregulated genes (*p* < 0.005)				
** *HRNR* **	hornerin	ENSG00000197915	12.71	6.87
** *RNF223* **	ring finger protein 223	ENSG00000237330	11.17	5.34
** *LCE3D* **	late cornified envelope 3D	ENSG00000163202	10.69	4.86
** *NLRP10* **	NLR family pyrin domain containing 10	ENSG00000182261	10.62	4.80
** *KLKP1* **	kallikrein pseudogene 1	ENSG00000197588	9.83	7.12
** *FGF22* **	fibroblast growth factor 22	ENSG00000070388	9.59	6.88
** *FAM25A* **	family with sequence similarity 25 member A	ENSG00000188100	8.91	3.14
** *KRT9* **	keratin 9	ENSG00000171403	6.85	9.18
** *ACP7* **	acid phosphatase 7, tartrate resistant (putative)	ENSG00000183760	6.16	5.92
** *CDSN* **	corneodesmosin	ENSG00000204539	5.69	7.09
downregulated genes (*p* < 0.005)				
** *DCD* **	dermcidin	ENSG00000161634	−14.45	11.83
** *SCGB2A2* **	secretoglobin family 2A member 2	ENSG00000110484	−14.15	8.31
** *PIP* **	prolactin induced protein	ENSG00000159763	−13.12	7.28
** *SCGB1D2* **	secretoglobin family 1D member 2	ENSG00000124935	−12.49	6.66
** *KRT19* **	keratin 19	ENSG00000171345	−10.85	5.03
** *MTND3P19* **	MT-ND3 pseudogene 19	ENSG00000271480	−10.78	4.97
** *PLAAT3* **	phospholipase A and acyltransferase 3	ENSG00000176485	−10.57	4.77
** *STAC2* **	SH3 and cysteine rich domain 2	ENSG00000141750	−10.48	4.68
** *ARFGEF3* **	ARFGEF family member 3	ENSG00000112379	−10.21	4.41
** *TP53I11* **	tumor protein p53 inducible protein 11	ENSG00000175274	−9.77	3.99
** *G0S2* **	G0/G1 switch 2	ENSG00000123689	−9.67	3.89
** *KCNK5* **	potassium two pore domain channel subfamily K member 5	ENSG00000164626	−9.67	3.89
** *PLIN1* **	perilipin 1	ENSG00000166819	−9.65	3.88
** *MRAS* **	muscle RAS oncogene homolog	ENSG00000158186	−9.63	3.85
** *DBN1* **	drebrin 1	ENSG00000113758	−9.57	3.80
** *SCN9A* **	sodium voltage-gated channel alpha subunit 9	ENSG00000169432	−9.55	3.78
** *GABRP* **	gamma-aminobutyric acid type A receptor subunit pi	ENSG00000094755	−9.54	3.77
** *CRACR2B* **	calcium release activated channel regulator 2B	ENSG00000177685	−9.54	3.77
** *ATP6V1B1* **	ATPase H+ transporting V1 subunit B1	ENSG00000116039	−9.52	3.75
** *PHYHIP* **	phytanoyl-CoA 2-hydroxylase interacting protein	ENSG00000168490	−9.46	3.69
** *MMP7* **	matrix metallopeptidase 7	ENSG00000137673	−9.36	3.60
** *TMEM200B* **	transmembrane protein 200B	ENSG00000253304	−9.36	3.60
** *TNFSF13B* **	TNF superfamily member 13b	ENSG00000102524	−9.29	3.53
** *TMEM213* **	transmembrane protein 213	ENSG00000214128	−9.24	3.48
** *ZNF891* **	zinc finger protein 891	ENSG00000214029	−9.22	3.47
** *ZNF419* **	zinc finger protein 419	ENSG00000105136	−9.20	3.45
** *EMP3* **	epithelial membrane protein 3	ENSG00000142227	−9.14	3.39
** *PNPLA6* **	patatin-like phospholipase domain containing 6	ENSG00000032444	−9.04	3.30
** *NNAT* **	neuronatin	ENSG00000053438	−9.02	3.28
** *PADI2* **	peptidyl arginine deiminase 2	ENSG00000117115	−9.02	3.28
** *RGMA* **	repulsive guidance molecule BMP co-receptor a	ENSG00000182175	−9.02	3.28
** *ESRRG* **	estrogen-related receptor gamma	ENSG00000196482	−9.00	3.26
** *PAMR1* **	peptidase domain containing associated with muscle regeneration 1	ENSG00000149090	−8.98	3.24
** *PODN* **	podocan	ENSG00000174348	−8.98	3.24
** *HLA-DQB2* **	major histocompatibility complex, class II, DQ beta 2	ENSG00000232629	−8.98	3.24
** *ADIPOQ* **	adiponectin, C1Q and collagen domain containing	ENSG00000181092	−8.93	3.19
** *CLDN7* **	claudin 7	ENSG00000181885	−8.93	3.19
** *ZNF528-AS1* **	ZNF528 antisense RNA 1	ENSG00000269834	−8.93	3.19
** *PPP1R1A* **	protein phosphatase 1 regulatory inhibitor subunit 1A	ENSG00000135447	−8.91	3.17
** *C7* **	complement C7	ENSG00000112936	−8.88	3.15
** *GPR12* **	G protein-coupled receptor 12	ENSG00000132975	−8.88	3.15
** *SMARCD3* **	SWI/SNF-related, matrix-associated, actin-dependent regulator of chromatin, subfamily d, member 3	ENSG00000082014	−8.84	3.11
** *CMTM7* **	CKLF-like MARVEL transmembrane domain containing 7	ENSG00000153551	−8.84	3.11
** *MUCL1* **	mucin-like 1	ENSG00000172551	−8.17	8.00
** *P4HA1* **	prolyl 4-hydroxylase subunit alpha 1	ENSG00000122884	−7.61	5.02
** *H19* **	H19 imprinted maternally expressed transcript	ENSG00000130600	−7.45	7.29
** *FABP4* **	fatty acid binding protein 4	ENSG00000170323	−7.28	5.61
** *CA6* **	carbonic anhydrase 6	ENSG00000131686	−7.15	4.58
** *C3* **	complement C3	ENSG00000125730	−6.77	5.65
** *PLIN4* **	perilipin 4	ENSG00000167676	−6.53	5.42
** *ST6GAL1* **	ST6 beta-galactoside alpha-2,6-sialyltransferase 1	ENSG00000073849	−6.52	3.97
** *SYNM* **	synemin	ENSG00000182253	−6.24	5.14
** *HLA-DRB5* **	major histocompatibility complex, class II, DR beta 5	ENSG00000198502	−5.73	6.58

**Table 2 ijms-24-09459-t002:** The upregulated and downregulated biological processes in KL lesion.

GOID	NodeSize	SampleMatch	Phyper	Padj	Term
UPREGULATED BIOLOGICAL PROCESSES					
GO:0009888	2271	41	4.67 × 10^−6^	0.001028	tissue development
GO:0060429	1428	31	2.31 × 10^−6^	0.000509	epithelium development
GO:0008544	535	30	4.47 × 10^−16^	9.83 × 10^−14^	epidermis development
GO:0043588	471	29	1.25 × 10^−16^	2.75 × 10^−14^	skin development
GO:0030855	858	29	3.97 × 10^−10^	8.74 × 10^−8^	epithelial cell differentiation
GO:0030216	349	25	5.82 × 10^−16^	1.28 × 10^−13^	keratinocyte differentiation
GO:0009913	417	25	3.50 × 10^−14^	7.69 × 10^−12^	epidermal cell differentiation
GO:0031424	269	22	2.37 × 10^−15^	5.22 × 10^−13^	keratinization
GO:0070268	126	16	1.88 × 10^−14^	4.13 × 10^−12^	cornification
GO:0006323	224	10	2.94 × 10^−5^	0.006477	dna packaging
GO:0018149	37	9	2.50 × 10^−11^	5.50 × 10^−9^	peptide cross-linking
GO:0006342	144	9	5.12 × 10^−6^	0.001127	chromatin silencing
GO:0019730	155	9	9.32 × 10^−6^	0.002051	antimicrobial humoral response
GO:0045814	163	9	1.40 × 10^−5^	0.003074	negative regulation of gene expression, epigenetic
GO:0050832	53	6	6.78 × 10^−6^	0.001492	defense response to fungus
GO:0009620	67	6	2.66 × 10^−5^	0.005861	response to fungus
GO:0043163	4	3	2.63 × 10^−6^	0.000578	cell envelope organization
GO:0045229	4	3	2.63 × 10^−6^	0.000578	external encapsulating structure organization
DOWNREGULATED BIOLOGICAL PROCESSES					
GO:0065007	14,167	862	1.18 × 10^−6^	0.002075	biological regulation
GO:0050789	13,392	817	4.64 × 10^−6^	0.008183	regulation of biological process
GO:0032502	6999	488	5.83 × 10^−10^	1.03 × 10^−6^	developmental process
GO:0048518	7074	471	1.05 × 10^−6^	0.001859	positive regulation of biological process
GO:0048856	6499	455	2.34 × 10^−9^	4.13 × 10^−6^	anatomical structure development
GO:0007275	5961	421	5.16 × 10^−9^	9.10 × 10^−6^	multicellular organism development
GO:0048731	5319	388	5.54 ×10^−10^	9.78 × 10^−7^	system development
GO:0048869	4874	348	8.92 × 10^−8^	0.000157	cellular developmental process
GO:0048583	5020	345	5.00 × 10^−6^	0.008828	regulation of response to stimulus
GO:0030154	4652	340	1.07 × 10^−8^	1.89 × 10^−5^	cell differentiation
GO:0065008	4501	318	1.23 × 10^−6^	0.002163	regulation of biological quality
GO:0048513	3899	295	5.41 × 10^−9^	9.55 × 10^−6^	animal organ development
GO:0023051	3941	295	1.63 × 10^−8^	2.87 × 10^−5^	regulation of signaling
GO:0010646	3891	289	5.25 × 10^−8^	9.27 × 10^−5^	regulation of cell communication
GO:0051239	3691	278	2.85 × 10^−8^	5.03 × 10^−5^	regulation of multicellular organismal process
GO:0009966	3481	256	9.51 × 10^−7^	0.001678	regulation of signal transduction
GO:0032879	3181	251	2.68 × 10^−9^	4.74 × 10^−6^	regulation of localization
GO:0009653	2981	241	6.57 × 10^−10^	1.16 × 10^−6^	anatomical structure morphogenesis
GO:0050793	3003	237	8.62 × 10^−9^	1.52 × 10^−5^	regulation of developmental process
GO:0007399	2556	195	2.66 × 10^−6^	0.004684	nervous system development
GO:0006928	2437	192	3.69 × 10^−7^	0.000652	movement of cell or subcellular component
GO:2000026	2340	185	5.02 × 10^−7^	0.000885	regulation of multicellular organismal development
GO:0048468	2303	182	6.51 × 10^−7^	0.001149	cell development
GO:0022610	1614	170	2.25 × 10^−16^	3.98 × 10^−13^	biological adhesion
GO:0007155	1607	170	1.49 × 10^−16^	2.63 × 10^−13^	cell adhesion
GO:0045595	2077	170	1.74 × 10^−7^	0.000308	regulation of cell differentiation
GO:0051049	2127	168	1.97 × 10^−6^	0.003482	regulation of transport
GO:0042592	2080	165	1.91 × 10^−6^	0.003374	homeostatic process
GO:0051240	2052	161	4.92 × 10^−6^	0.008674	positive regulation of multicellular organismal process
GO:0006811	1810	157	1.36 × 10^−8^	2.40 × 10^−5^	ion transport
GO:0048870	1949	155	3.57 × 10^−6^	0.006306	cell motility
GO:0051674	1949	155	3.57 × 10^−6^	0.006306	localization of cell
GO:0007267	1811	146	3.43 × 10^−6^	0.006052	cell-cell signaling
GO:0016477	1780	144	3.35 × 10^−6^	0.005909	cell migration
GO:0022008	1756	142	4.04 × 10^−6^	0.007127	neurogenesis
GO:0051094	1558	137	5.82 × 10^−8^	0.000103	positive regulation of developmental process
GO:0006812	1276	115	2.16 × 10^−7^	0.000382	cation transport
GO:0048878	1286	113	9.80 × 10^−7^	0.001729	chemical homeostasis
GO:0048646	1288	110	5.15 × 10^−6^	0.009087	anatomical structure formation involved in morphogenesis
GO:0035295	1199	109	3.01 × 10^−7^	0.00053	tube development
GO:0051270	1198	106	1.58 × 10^−6^	0.002793	regulation of cellular component movement
GO:0045597	1091	105	3.00 × 10^−8^	5.29 × 10^−5^	positive regulation of cell differentiation
GO:0009887	1142	101	2.89 × 10^−6^	0.005093	animal organ morphogenesis
GO:0019725	1052	99	2.23 × 10^−7^	0.000393	cellular homeostasis
GO:2000145	1110	98	4.40 × 10^−6^	0.007754	regulation of cell motility
GO:0030001	978	96	4.77 × 10^−8^	8.42 × 10^−5^	metal ion transport
GO:0098609	983	95	1.17 × 10^−7^	0.000207	cell-cell adhesion
GO:0035239	994	93	6.83 × 10^−7^	0.001205	tube morphogenesis
GO:0055082	890	89	6.55 × 10^−8^	0.000115	cellular chemical homeostasis
GO:0050801	868	88	4.18 × 10^−8^	7.37 × 10^−5^	ion homeostasis
GO:0030155	839	84	1.48 × 10^−7^	0.000261	regulation of cell adhesion
GO:0098771	796	82	6.20 × 10^−8^	0.000109	inorganic ion homeostasis
GO:0043269	754	80	2.54 × 10^−8^	4.48 × 10^−5^	regulation of ion transport
GO:0055080	784	79	2.71 × 10^−7^	0.000478	cation homeostasis
GO:0006873	722	79	8.09 × 10^−9^	1.43 × 10^−5^	cellular ion homeostasis
GO:0030003	708	77	1.64 × 10^−8^	2.89 × 10^−5^	cellular cation homeostasis
GO:0055065	696	72	3.44 × 10^−7^	0.000608	metal ion homeostasis
GO:0006875	624	69	4.86 × 10^−8^	8.57 × 10^−5^	cellular metal ion homeostasis
GO:0006935	703	69	3.92 × 10^−6^	0.006915	chemotaxis
GO:0042330	705	69	4.32 × 10^−6^	0.007623	taxis
GO:0048598	631	67	3.46 × 10^−7^	0.00061	embryonic morphogenesis
GO:0072507	553	63	6.35 × 10^−8^	0.000112	divalent inorganic cation homeostasis
GO:0034762	602	63	1.27 × 10^−6^	0.002245	regulation of transmembrane transport
GO:0055074	507	62	5.71 × 10^−9^	1.01 × 10^−5^	calcium ion homeostasis
GO:0072503	534	61	9.34 × 10^−8^	0.000165	cellular divalent inorganic cation homeostasis
GO:0002521	581	61	1.69 × 10^−6^	0.002975	leukocyte differentiation
GO:0006874	495	60	1.40 × 10^−8^	2.47 × 10^−5^	cellular calcium ion homeostasis
GO:0045785	502	54	3.24 × 10^−6^	0.005708	positive regulation of cell adhesion
GO:0010959	437	52	2.34 × 10^−7^	0.000412	regulation of metal ion transport
GO:0031589	377	51	4.63 × 10^−9^	8.17 × 10^−6^	cell-substrate adhesion
GO:0051480	386	48	1.83 × 10^−7^	0.000323	regulation of cytosolic calcium ion concentration
GO:0030198	404	47	1.67 × 10^−6^	0.002948	extracellular matrix organization
GO:0007204	347	43	8.69 × 10^−7^	0.001532	positive regulation of cytosolic calcium ion concentration
GO:0007160	249	34	1.37 × 10^−6^	0.002416	cell-matrix adhesion
GO:0050920	234	32	2.69 × 10^−6^	0.004737	regulation of chemotaxis
GO:0060402	181	27	3.06 × 10^−6^	0.005396	calcium ion transport into cytosol
GO:0051209	144	23	5.07 × 10^−6^	0.008948	release of sequestered calcium ion into cytosol
GO:0002407	25	9	4.71 × 10^−6^	0.008312	dendritic cell chemotaxis
GO:0071313	10	6	5.24 × 10^−6^	0.009242	cellular response to caffeine

## Data Availability

The data presented in this study are available on request from the corresponding author. The data are not publicly available due to ethical reasons.

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
