# Peer review of "The Epidermal Transcriptome Analysis of a Novel c.639_642dup LORICRIN Variant-Delineation of the Loricrin Keratoderma Pathology"

_ijms, 2023, doi:10.3390/ijms24119459_

Round 1
Reviewer 1 Report
The article entitled «The epidermal transcriptome analysis of a novel c.639_642dup LORICRIN variant - delineation of the loricrin keratoderma pathology” by Katarzyna Wertheim-Tysarowska et al is an interesting description of 2 mutations in the Loricrin genes, one leading to the development of ichthyosis and palmoplantar keratoderma.
Mutations of Loricrin gene are rare and only little information is available, therefore molecular data reported in this paper could be considered as a good beginning for further investigations.
Comments:
I understand that RNAseq was performed only in one LK patient with c.639_642dup 2 whereas, apparently 3 members of the same family were known (2 sisters and father). Testing the 3 patients would have been more relevant statistically.
It seems that RNAseq was performed in only one biopsy on the upper tibia, whereas the ichthyosis was diffuse and the keratoderma was palmoplantar. What was the reason for the choice ? Precise if a replicate of RNAseq has been performed ?
By the way has the expression of HRNR already been evaluated in other skin diseases with keratodermas ?
Clinically it would be interesting to indicate if the keratoderma of these patients is painful and inflammatory, as it is the case in other types of keratodermas. It would give some directions in comparison to other keratodermas for the analysis of RNAseq data. By the way have the authors looked at the existence of RNAseq data in other types of palmoplantar keratodermas (Pachyonychia Congenita, Olmsted or EBS Dowling-Meara) ?
Abbreviations should be explained when they first appear in the text (there are only few in the transcriptome analysis), for instance DEG, FC...
Also check the typography please,
For instance: lines 224-229 “Since, our results are, to our knowledge, the first transcriptome analysis of LK lesion and were performed on the one patient only, replicative studies are needed. Nevertheless, this results give novel -insight into the pathogenesis of the disease and may have therapeutic implications in future.
Another issue raised by u,s concerns the clinical significance of nonsense variants in the LORICRIN
Reviewer 2 Report
The main question addressed by the research is the transcriptome profiles of lesional loricrin keratoderma epidermis of a patient with a novel gene mutation.As a result of transcriptome analysis in this study, it was revealed that the expression of the hornerin gene was elevated in the affected areas. This suggested the importance of hornerin in the epidermis, whose role had not been well understood until then, and laid the groundwork for future research.
No clinical photographs of Family 1 are shown in this manuscript. The presence or absence of the honeycomb pattern or pseudoainhum is also not stated. This reviewer believed that these issues should be resolved.
Author Response
The authors would like to thank the Reviewer for the revision of the text and constructive remarks. Please find the answer to the remarks below.
- No clinical photographs of Family 1 are shown in this manuscript. The presence or absence of the honeycomb pattern or pseudoainhum is also not stated. This reviewer believed that these issues should be resolved.
We fully agree with the Reviewer, that the clinical photographs would be valuable. Unfortunately, despite our greatest attempts, the family did not agree to publish any pictures of them, including those presenting only palm/ tibia etc. Therefore, according to Reviewer’s suggestion, we broaden the clinical synopsis in the text and included information about honeycomb pattern of PPK and pseudoainhum.
Round 2
Reviewer 2 Report
The authors made appropriate modifications to the manuscript.